# Vitamin D Associated with Exercise Can Be Used as a Promising Tool in Neurodegenerative Disease Protection

**DOI:** 10.3390/molecules30183823

**Published:** 2025-09-21

**Authors:** Gabriele Farina, Clara Crescioli

**Affiliations:** Department of Movement, Human and Health Sciences, University of Rome Foro Italico, 00135 Rome, Italy; g.farina2@studenti.uniroma4.it

**Keywords:** vitamin D, neuroinflammation, neurodegeneration, exercise

## Abstract

Neurodegenerative diseases, including Parkinson’s disease, Alzheimer’s disease, multiple sclerosis, and Huntington’s disease, represent unmet medical and social needs. Still, no definitive cure exists for these illnesses, hence a therapeutic approach with molecules able to prevent/downtone/modify the disease seems highly attractive. Remarkably, a higher risk of neurodegenerative disease is associated with low vitamin D levels. Vitamin D is a multifaceted molecule able to target critical neuroinflammatory processes underlying neurodegeneration, acting through genomic or rapid signaling. This narrative review aims to focus on vitamin D’s potential to be an optimal neuroprotective molecule, based on its ability to target and counteract aberrant biomolecular processes involved in neuroinflammation/neurodegeneration. Noticeably, exercise can potentiate vitamin D’s protective effect through some anti-inflammatory actions exerted on shared biomolecular targets. Thus, although vitamin D is not strictly a drug, it could be potentially allocated within the therapeutic approach to neurodegenerative diseases in combination with adapted exercise, best as an early intervention. Topics on the complexity concerning the doses for supplementation and data discrepancy from trials are addressed. The urgent demand is to test and clarify vitamin D efficacy and safety, combined or not with exercise, in clinical settings.

## 1. Introduction

Vitamin D-dependent regulation goes beyond classical bone phospho-calcium homeostasis and extends to other organ or tissue functions, including the central nervous system (CNS) [1]. Currently, vitamin D pleiotropic actions are acknowledged due to its ability to modulate many biological processes, including cell proliferation and differentiation, immune response, and inflammation [1].

Acting through the vitamin D receptor (VDR), which is expressed in almost all brain areas, vitamin D can behave as a neuroprotective molecule since it inhibits a variety of processes, i.e., oxidative stress, inflammation, and pathogenic protein formation and aggregation, which underlie and precede neurodegeneration [2]. In humans, vitamin D insufficiency increases the susceptibility to neurodegenerative diseases, such as Parkinson’s disease (PD), Alzheimer’s disease (AD), multiple sclerosis (MS), Huntington’s disease (HD), cognitive disorders, and dementia, although causative links are still to be fully elucidated [3]. Furthermore, lower vitamin D levels associate with higher neurodegenerative stages, whereas vitamin D sufficiency, i.e., reached after supplementation, seems to contribute to a better prognosis [4].

Currently, half of the people across the world exhibit poor vitamin D levels, with more than one billion people suffering from vitamin D deficiency—defined as less than 30 nmol/L (<12 ng/mL) [5,6].

Furthermore, the neurodegenerative occurrence, meaning both incidence and prevalence, increases with the increasing number of elderly people, representing a growing challenge in public health.

The aim of this narrative review is to focus on the potential neuroprotective role of vitamin D, which exhibits higher efficacy when combined with physical exercise (PE), since this molecule and PE share some neuroinflammatory/neurodegenerative pathways as common targets. Neuroinflammation and neurodegeneration are addressed as interacting parts of a vicious loop that needs to be counteracted as early as possible to control disease development. Based on its anti-inflammatory properties targeting the vicious loop, vitamin D is suggested as a complementary molecule to be allocated in therapeutic options for neurodegenerative diseases to ameliorate prevention, management, and outcome. Importantly, the synergistic effect between vitamin D and PE is discussed as an emerging attractive issue to focus on, hopefully suggesting new scenarios in a multidimensional approach to neurogenerative diseases.

## 2. The Versatile Identity of Vitamin D

This vitamin, classified as “D” since it is the fourth to be discovered, although belonging to the vitamin class, functions as a prototypical secosteroid hormone (a steroid subclass with a broken ring). In humans, the majority of vitamin D_3_, or cholecalciferol, originates mainly in skin from enzymatic conversion of the precursor 7-dehydrocholesterol under ultraviolet (UV) rays and only 20% from diet [7]. Although the 25(OH)D is the more abundant and stable metabolite in circulation—used, in fact, for vitamin D serum level determination—the 1.25(OH2)D metabolite retains 500 times higher affinity for VDR [8].

Upon VDR binding, vitamin D signals via a hormone-like classic mechanism, heterodimerizing with 9-cis-retinoic acid receptor X receptor (RXR) to form the dimeric complex VDR:RXR, which targets the promoter regions within vitamin D-responsive elements (VDREs) and starts gene transcription [9,10]. Vitamin D can also signal via a non-genomic rapid mechanism, in which the VDR translocates from cytoplasm to plasma membrane through caveolae, which are plasmalemma microdomains highly specialized for macromolecule transcytosis, and activates transmembrane rapid signal transduction paths within seconds to minutes [11].

A membrane-associated rapid-response steroid-binding protein (MARRS) dedicated to vitamin D non-genomic effects is described in [12]. Through these mechanisms, vitamin D exerts a fine-tuned regulation on an exceptionally broad spectrum of biological activities, highly impacting human health.

Indeed, besides the classical regulation of calcium-phosphorus for bone homeostasis, vitamin D status regulates several biological processes, i.e., energy metabolism and skeletal muscle function. Importantly, vitamin D drives the immune response to an anti-inflammatory profile, i.e., downregulating T helper (Th)1/Th17 phenotypes, both deeply involved in inflammatory overresponse [13,14], and shifting human brain microglial from proinflammatory M1 to anti-inflammatory M2 phenotype [15]. Hence, considering its anti-inflammatory role, vitamin D has gained high interest as a neuroprotective molecule.

Although neuroinflammation is usually described as the cause preceding neurodegeneration, these processes occur almost simultaneously and establish a mutual detrimental dialogue, as addressed in the following paragraph.

## 3. Neuroinflammation and Neurodegeneration: A Two-Way Route

The question of whether inflammation is the cause or consequence of neurodegeneration is still debated. Like the whole body, inflammation in the CNS is a physiological immune response to harmful stimuli, i.e., traumas or infections, behaving as a safeguard of the cell microenvironment [16]. CNS resident immune cells, the microglia, represent the main immune defense in the brain, which, under challenge, trigger a defensive neuroinflammatory response characterized by cellular and signaling overresponse, dysfunction of the brain–blood barrier (BBB), release of inflammatory mediators, such as reactive oxygen species (ROS), nitrogen species, cytokines, and chemokines [17,18].

Whenever this process is inappropriately protracted, a chronic neuroinflammatory state occurs and potentially represents the first step toward different neurodegenerative diseases, independently of their specific characteristics [17,19].

### The Systemic-Local Detrimental Loop in CNS

The causes inducing chronic neuroinflammation are numerous and still far from being well elucidated. The immunocyte infiltration into CNS tissue from the peripheral areas likely occurs due to a consequence of a robust inflammatory peripheral response, triggered, i.e., by systemic exposure to inflammation inducers, such as lipopolysaccharides, or viruses [20]. This process would first prompt the activation of microglia to produce and release into the BBB inflammatory mediators, which, in turn, recall from the peripheral immune system additional infiltration of immune cells, such as T cells and macrophages, further weakening the barrier [21,22]. Thus, a vicious circle starts.

The activated microglia, astrocytes (a subtype of glial cells with homeostatic function), oligodendrocytes, pericytes, endothelial cells, and peripheral immunocytes interconnect and establish a complex web of proinflammatory molecules and aberrant signaling, contributing to inflammatory microenvironment maintenance and tissue injury [23,24]. These processes converge in a self-enhancing, detrimental circle between systemic and local areas.

In this scenario, aberrant signaling cascades engaging pattern recognition receptors (PRRs), such as toll-like receptors (TLRs), receptors for advanced glycation end (RAGE) products, nod-like receptor protein (NLRP)3 inflammasome, damage-associated molecular pattern (DAMP) profile, accumulation of reactive oxygen species (ROS), cytokines and chemokines, protein aggregates, adenosine triphosphate (ATP), and mitochondrial DNA released by damaged neurons, act as inflammation enhancers, as exhaustively described elsewhere [25].

Furthermore, a huge release of proinflammatory cytokines, including tumor necrosis factor (TNFα), interleukin (IL)-6, IL-23, IL-1β, interferon (IFN)γ, and granulocyte macrophage colony-stimulating factor (GM-CSF), further sustains and contributes to disease progression. As examples, IL-23, IL-6, or IL-1β, produced by CNS resident immunocytes, amplify T cell pathogenic activity in neuroinflammation, and GM-CSF promotes tissue damage through macrophage infiltration [26,27]. Noticeably, these cytokines are targets of vitamin D, as addressed in the following paragraph.

The large cytokine release, once activated, induces anomalous signaling cascades, which, in turn, trigger synaptic/neuronal loss and aberrant protein release/accumulation, which ends in plaque formation and worsens neuronal dysfunction, such as amyloid beta (Aβ) in AD [28]. Indeed, brain tissues and blood from AD patients show high levels of inflammatory cytokines derived from microglia, which, failing to efficiently remove Aβ, become chronically activated and intensify protein aggregation or plaque formation.

Similarly, in PD, the aberrant deposition of α-synuclein ends in aggregates (oligomers or fibrils), which further activate microglia [29]. The excess of superoxide, nitric oxide, and TNFα derived from chronically activated microglia culminates in neurotoxic effects, typical of the disease. Furthermore, the CNSs of PD patients show massive CD4+ and CD8+ T cell infiltration, which likely precedes pathological α-synuclein deposition [17]. The mechanisms underlying PD development are intricate and still not fully elucidated. Moreover, the gut–brain axis seems to contribute to the disease progression via gut dysbiosis-dependent inflammatory cascade and BBB permeability modification [30]. The latter gives permission for extra cell infiltration. Hence, the inflammatory loop is perpetuated and exhibits a mutual-promotion pattern to aggravate neurodegeneration, acknowledged as a pathological condition common to several neurodegenerative diseases exhibiting different pathogenetic mechanisms, protein aggregates, and genetic variations [19]. Figure 1 summarizes the link between neuroinflammation and neurodegeneration.

Undeniably, limiting the neuroinflammatory loop as soon as possible, ideally before clinical signs, would be of pivotal importance.

The current therapeutic approaches to neurodegenerative diseases such as AD, PD, HD, or MS include anti-inflammatory and antioxidant drugs, to limit inflammation, immune infiltration, and metabolic failure in CNS; molecules regulating myelin deposition and increasing autophagic flux in neurons; drugs enabling amyloid clearance; molecules targeting transcriptional regulation or epigenetic mechanisms, aimed at modulating widespread downstream effects [31].

To date, most of these drugs show promising results in experimental models, often not immediately translated to humans, showing, indeed, controversial results between preclinical models and trials. The discrepancy might also be due to species-specific features.

In addition, data from trials, whose primary outcome is aimed at targeting neurodegeneration-associated protein aggregates, are too often disappointing because the cognitive function is not reestablished, even when aggregates are eliminated by the treatment [32].

In this scenario, there is growing evidence and interest in vitamin D status that can contribute to brain health or disease, depending on sufficient or deficient levels, respectively.

## 4. Vitamin D Status: Towards Neuroprotection or Neurodegeneration

The putative role of vitamin D as a neuroprotectant mainly relies on its multitargeting effects within the CNS, since this molecule can target neuronal cells and non-neural cells, such as endothelial cells involved in neurovascular dysfunction [33,34,35].

Upon binding to the VDR, which is ubiquitously expressed in human brain areas—hypothalamus, hippocampal formation, amygdala, stria terminalis, and neocortex [36,37,38], vitamin D can substantially counteract homeostasis loss both with genomic and non-genomic mechanisms, exerting pro-survival/anti-inflammatory effects to prevent neuronal cell loss or demyelination [39,40,41].

Vitamin D-deficient status seems to contribute to the progression of neurodegeneration via Wnt signaling failure [42]. Interestingly, Wnt signaling is highly conserved and it is known to exert a critical role in synaptic maintenance and neuronal functions, acting either through canonical or non-canonical paths, whose deregulation is found in AD [43].

Vitamin D is documented to maintain the Wnt pathway to warrant brain plasticity and promote brain circuit regulation, both in early and late life [44]. Accordingly, a reduced expression of VDR impairs Wnt-dependent cascade affecting neural cell signaling and unbalancing the neurogenic homeostasis, such as in cognitive impairments, as observed in hippocampal cells of old brains [42].

It seems that hypovitaminosis D can promote cognitive decline, reducing neurogenesis by upregulating Wnt inhibitors, such as Dickkopf-1 (DKK1); conversely, adequate vitamin D levels sustain Wnt signaling, repressing its inhibitors. This latter effect likely prolongs the time of stem cell quiescence, preserving the staminal reservoir [45]. So far, in neural cells, vitamin D seems to act as a co-activator of Wnt, whereas in other cell types, i.e., cancer cells, this molecule acts as a Wnt antagonist [46]. To date, the complex interaction between vitamin D and Wnt is dependent on the microenvironment and not fully elucidated.

The risk of PD development seems to be closely associated with vitamin D deficiency and vitamin D–VDR axis failure [47]. Indeed, VDR-regulated genes, such as nuclear receptor-related 1 protein (Nurr1) gene (critical in dopaminergic system development and maintenance) and tyrosine hydroxylase (which catalyzes L-DOPA formation), are likely co-responsible for pathomechanisms of PD [48,49]. Within the nigrostriatal signaling, the substantia nigra, the area deeply involved in dopaminergic neuron degeneration, highly expresses both VDR and 1α-hydroxylase, specific for the enzymatic conversion to vitamin D’s active form, and it is likely that a deficient vitamin D/VDR system accelerates the disease [50]. Experimental in vitro studies document that the addition of calcitriol can restore pro-survival cell signaling, such as phosphatidylinositol 3-kinase (PI3K) pathway, and increase glial cell-derived neurotrophic factor (GDNF) [51,52]. Consequently, cell viability and proliferation are both improved, and oxidative stress/ROS production, α-synuclein aggregation, and neurotoxicity are reduced [53].

Remarkably, restoring the function of the vitamin D-VDR axis can inhibit aberrancies in NLRP3/caspase1 signaling, mitigate inflammation and membrane permeabilization, significantly attenuating microglia overactivation, as recently shown in a PD experimental study [54]. Nevertheless, there is evidence that while PD risk increases with D deficiency, vitamin D supplementation plays little role, especially in well-established disease [55].

Indeed, upon VDR binding, vitamin D significantly reduces the level of proinflammatory cytokines such as TNFα, IL-1β, IL-6, deeply involved in neuroinflammatory and pathogenetic mechanisms in PD, AD, or MS neurodegeneration, and protects neural tissues from damage through the increase in anti-inflammatory cytokine IL-10 [56,57]. Vitamin D-induced IL-10 increase counteracts inflammation through induction of suppressor of cytokine signaling 3 (SOCS3) [58]. In the PD mouse model, vitamin D has been shown to increase other anti-inflammatory molecules, such as transforming growth factor (TGF)β, IL-4, and cluster of differentiation (CD)163, CD202, and CD206, involved against oxidative damage and in the shift to a protolerogenic profile [59].

Furthermore, vitamin D can modulate tissue architecture through extracellular matrix deposition and fibrosis development, targeting the enzymatic activity of metalloproteinases (MMP) and TGFβ [41,60].

Vitamin D is known to modulate mitochondrial function and reduce neuronal apoptosis via AMP-activated protein kinase/protein kinase B/glycogen synthase kinase (AMPK/AKT/GSK)-3β signaling path activation, displaying, therefore, a reparatory activity besides the neurogenic effect [61,62].

Vitamin D-induced maintenance of mitochondrial function entails ROS reduction and regulates antioxidant expression, on one side, via vitamin D-Klotho-nuclear factor erythroid 2-related factor 2 (Nrf2) regulatory network, glutathione, and superoxide dismutase expression upregulation, on the other side, via nitric oxide (NO)/inducible nitric oxide synthase (iNOS) downregulation [63,64,65,66].

Altogether, the anti-inflammatory/antioxidant/trophic effects of vitamin D undeniably converge in maintaining as intact as possible the synaptic plasticity, cell cytoskeleton, and molecular transport, whereas vitamin D deficiency results in loss of function [67,68,69].

In line with in vitro data, results from experimental models show that vitamin D intake in deficient status reduces oxidative stress and dopaminergic loss, restores endothelial function, and improves neurogenesis and working memory, highlighting the importance of administration timing, with the earliest intervention being the most efficient, as shown in PD and AD [70,71,72,73,74]. Of note, the observation on early administration timing is in line with the concept of limiting the neuroinflammatory loop as soon as possible, as previously addressed.

In PD patients, vitamin D can restore motor functionality and rebalance the immune homeostasis [75].

Vitamin D can mitigate the Th1 proinflammatory effects, significantly reducing both circulating and local Th1-type cytokines and chemokines, such as interferon gamma (IFNγ), TNFα, IL-1, IL-2, IL-6, IL-1β, and C-C or C-X-C motif chemokines [76,77]. This cytokine inhibition is mostly due to nuclear factor-kB (NF-kB) p65 block via the upregulation of the inhibitory protein inhibitor of kappa B-alpha (IkBa) [78].

This vitamin D preserves vascular cells from injury mainly through local reduction of intercellular adhesion molecule 1 (ICAM-1), vascular cell adhesion molecule 1 (VCAM-1), cyclooxygenase 2 (COX-2), E-selectin, or vascular tissue expression of MMP-2 and MMP-9 [41,79].

Thus far, vitamin D-induced neuroprotection is exerted at the systemic and local levels. Inhibiting inflammatory molecules at the cell/tissue level seems highly intriguing since this effect potentially interrupts the self-detrimental loop between systemic and local compartments, which aggravates neurodegeneration.

We have previously reported on VDR agonists as tools for the resolution of inflammation, directly targeting different human cell types, i.e., T cells, cardiac and muscular cells, prostate and bladder cells, and renal cells activated by a maximal inflammatory challenge [80,81,82,83,84]. The inhibition of the release of chemotactic mediators is a pivotal step to interrupt the vicious inflammatory circle.

Some of the vitamin D-induced local and systemic effects are reported in Figure 2.

Studies on humans and animals on HD document that vitamin D deficiency likely directly correlates with motor abnormalities, whereas vitamin D supplementation leads to clinical symptom improvement [85,86,87].

Furthermore, HD is reported to be associated with bone metabolism alteration, showing significantly lower bone mineral density and disease-specific osteoporosis, likely related to the number of CAG repeats—dominantly inherited CAG trinucleotide repeat in the huntingtin gene—that represents the most frequently evaluated marker of disease severity [88,89,90].

These relevant issues contribute to impairment of independent ambulation in HD, and, consequently, to falls, autonomy loss, and transfer to nursing homes. Nevertheless, there are no specific dedicated studies, which are potentially of great interest. To date, epidemiological or clinical studies focusing on the relationship between vitamin D status and HD are lacking, except for the explorative study from Chel, reporting a positive inverse association between functional impairment and vitamin D level [85]. In vitro studies show that vitamin D induces the neurotrophins nerve growth factor (NGF) and brain-derived neurotrophic factor (BDNF) in several neuronal cells, including striatal neurons, the main affected cell population in HD [91,92].

Some investigators report a relationship between reduced vitamin D blood level and higher risk of MS; accordingly, MS patients taking vitamin D show reduced neuroinflammation and neurodegeneration, likely due to an increase in tissue oxygenation [93,94]. As per data in the MS murine model, proteolipid protein (PLP), myelin basic protein (MBP), myelin oligodendrocyte glycoprotein (MOG), and CNP increase after vitamin D intake [95,96]. Levels of two isoforms of vitamin D binding proteins and apolipoprotein E in the cerebrospinal fluid are suggested as potential biomarkers for MS diagnosis, albeit some other studies evidence no relationship between vitamin D level and MS etiopathogenesis [97,98,99]. Thus, the need for further investigations into vitamin D therapeutic activity in MS is undeniable.

Similarly, few studies have been carried out on vitamin D in amyotrophic lateral sclerosis (ALS). ALS development seems associated with vitamin D status via the regulation of immune components, involving toll-like receptors (TLR), major histocompatibility complex (MHC) class II molecules, poly (ADP-ribose) polymerase 1 (PARP1), and heme oxygenase-1 (HO-1) [100]. Furthermore, vitamin D status affects ALS through the direct control of important cell signaling mechanisms, i.e., mitogen-activated protein kinase (MAPK), ROS, Wnt/β-catenin, MMP, and nitric oxidase synthase [100]. It should be underlined that, although vitamin D supplementation can improve motor functional capacity and decrease muscle weakness, and, therefore, improve the quality of life, it cannot modify disease prognosis or outcome [101]. Thus far, it must be underlined once more that the importance of further investigations on the potential role of vitamin D in ALS pathophysiology. However, the advantage of using vitamin D to supplement deficient/insufficient status appears undeniably as an optimal approach. Nevertheless, this issue opens a window on another important question: what is the best dose for supplementation?

It is acknowledged that the vitamin D dose may vary according to the individual’s need, which depends on several factors, including diet, sunlight exposure, sex, latitude, lifestyle, and inadequate renal conversion or intestinal absorption [102,103,104]. Thus, supplementation doses should be calibrated in response to skeletal and extra-skeletal needs.

According to guidelines, a 25(OH)D sufficient serum concentration should be 50–100 nmol/L; the Central European guideline recommends 75–125 nmol/L [104,105,106]. To date, the latest report, “Consensus Statement on vitamin D Status Assessment and Supplementation: Whys, Whens, and Hows,” evidences that the optimal 25(OH)D level remains to be clarified, particularly for extra-skeletal needs [107]. Concerning the effect of vitamin D supplementation in neurodegenerative diseases, data are conflicting as well. Trials on vitamin D supplementation in AD, PD, or MS patients report either no effect or some benefits on cognitive function, independently of the taken dose (from 800 IU/day to 1000 IU/day, or 6000 IU/day), as recently summarized [3]. It is mandatory to underline that the lack of reliable results from the trials also depends on the lack of assay standardization, high variability in methods, and, consequently, a non-homogeneous approach for vitamin D supplementation (different forms, metabolites, and doses) [5,6,107].

Remarkably, vitamin D’s effect on the CNS can be affected by sedentary or active behavior. This topic deserves particular attention, since physical exercise (PE) can affect, per se, the risk of neurodegenerative diseases and could exert a multiplicative effect when combined with vitamin D [108,109], as addressed in the following paragraph.

## 5. Vitamin D Status and Exercise: A Possible Synergy in Neuroprotection

There is growing evidence of some beneficial effects of PE against neurodegenerative disorders through several simultaneous actions [108].

PE is defined as designed, organized, and repeated aerobic/anaerobic physical activity characterized by defined frequency, duration, and intensity, with the specific goal of enhancing or maintaining physical fitness [110].

Interestingly, engaging in PE on a regular basis may slow aging-related cognitive decline [111,112]. An acknowledged effect of PE is the improvement of blood flow and, consequently, cerebral flow, which supports neuronal plasticity.

Clinical research shows that PE leads to increases in gray matter volume, particularly in the frontal cortex and hippocampus [113] and elevates the level of neurotrophic factors, such as peripheral BDNF [114].

Nowadays, adapted PE is fully recognized as an anti-inflammatory intervention since it significantly reduces the level of proinflammatory cytokines, including the cytokines resident in the CNS [115,116]. This effect is undeniably due to the direct regulation of PE onto immune cells, which is noted as “exercise immunology” [117,118,119] but also depends on the biomolecules released during skeletal muscle working, named myokines, which all exert anti-inflammatory and regulatory effects onto several body functions, acting within a network established among almost all body tissues and organs [120,121,122,123]. More than 600 have been found, although their functions are still to be clarified [124]. There are neurotrophic/anti-inflammatory/regulatory myokines able to cross the BBB and involved in muscle–brain cross-talk, i.e., BDNF, IL-6, irisin, cathepsin B, or muscle-derived insulin-like growth factor 1 (IGF-1), as previously reported, although their fine-tuned mechanisms are not yet elucidated [125,126].

Remarkably, recent evidence suggests a possible synergy between PE and vitamin D to counteract neuroinflammation through a merged action on several common targets.

To date, most of the myokines able to cross the BBB are sustained by vitamin D. As examples, vitamin D, through BDNF modulation, mitigates molecular derangements, improves neuronal outcomes in animal models with neurogeneration, and might support neuroprotection in vitamin D-deficient subjects (<30 ng/mL) [127,128]. Furthermore, higher BDNF levels are associated with the modulation of microglial activation by triggering a shift from the proinflammatory M1 phenotype to the anti-inflammatory and neuroprotective M2 phenotype, which, in turn, downregulates the secretion of inflammatory cytokines, including IL-1β and TNFα [129,130].

Muscle-derived irisin and IL-6 are increased by vitamin D level or VDR expression [131,132,133]; vitamin D upregulates cathepsin B expression and positively correlates with IGF-1 [134,135].

Another example comes from the observation that regular exercise can enhance VDR expression in the brain, particularly in regions essential for cognitive, memory, and emotional regulation, such as the hippocampus and prefrontal cortex [109,136]. This effect can trigger a virtuous neuroprotective loop since the increased VDR expression improves neuronal sensitivity to vitamin D, therefore enhancing PE-induced effects.

In addition, both PE and vitamin D target NF-κB, the central regulator of inflammatory responses, to inhibit the production of proinflammatory genes, such as TNFα [137]. Another example of targets shared by PE and vitamin D comes from the activation of the Nrf2 pathway, which plays a crucial role in antioxidant defense. Of note, Nrf2 activation increases the expression of enzymes such as superoxide dismutase, glutathione peroxidase, and catalase, and reduces oxidative stress and neuroinflammation [138,139].

Vitamin D exerts similar anti-inflammatory and antioxidant effects by activating the Nrf2 signaling pathway, by enhancing the expression of antioxidant genes such as heme oxygenase-1 (H-1) and glutathione peroxidase 4 (GPX4), and reducing ferroptosis in aging brains, all processes contributing to the reduction of neuroinflammation and CNS oxidative damage [139].

So far, these common molecular targets support the hypothesis of potential synergistic effects between vitamin D and exercise, which is useful for the prevention and treatment of neurodegenerative diseases. From a recent paper reporting data in an experimental model, the combination of exercise training and vitamin D results in a synergistic effect on brain health by improving cognitive function, BBB integrity, and mitochondrial efficiency [140].

Table 1 shows the common targets and pathways modulated by PE and vitamin D, together with the induced effects that may merge in a synergistic action.

Noticeably, it is generally acknowledged that PE increases testosterone in men depending on several factors, including types of exercise (endurance/resistance), training intensity and recovery times, and population (sedentary/active/athletes; young/old) [141]. PE can also induce the expression of androgen receptor (AR), resulting in general anti-inflammatory effects through STAT3, counteracting STAT3-induced sarcopenia [142]. Nowadays, androgens’ function is acknowledged to extend far beyond sexual function, reaching into the human brain to sustain neuroprotection, memory, and mitigate cognitive decline [143]. Indeed, androgens can directly modulate neuroplasticity, particularly in the hippocampus, and protect neurons from Aβ toxicity, involving several intracellular signaling cascades, such as extracellular signal-regulated kinase-cAMP response element-binding protein (ERK-CREB), MAPK/ERK, or protein tyrosine kinase (PTK) [143,144,145,146].

Androgen administration is hypothesized as a therapy for neurodegenerative diseases [147].

Whether or not vitamin D deficiency can impact testosterone levels in humans is still debated, as some studies reported a positive correlation between the levels of these hormones, while others did not, highlighting the need for further well-designed, long-term trials [148].

The research in humans on neuroprotection combining vitamin D and exercise and all related effects is still in its infancy and undeniably needs further investigations, both in vitro and in vivo.

## 6. Conclusions

Thus far, it is undeniable that vitamin D positively interferes with signal paths engaged in the pathogenesis of neurodegenerative diseases, i.e., targeting the cellular signals contributing to neuronal loss or abnormal protein aggregates, which represent the common features of neurodegenerative diseases. Indeed, vitamin D-induced multifaceted actions, such as anti-inflammatory, antioxidative, immunomodulatory, neurovascular, and trophic effects, merge and contribute to neuroprotection.

At the same time, the discrepancy reported in results from clinical studies makes the research on this topic jump a step back. To date, concerns about the conflicting data from clinical trials reflect the inconsistency of the methods used, including dose and duration variability of vitamin D treatment. Indeed, investigations are mandatory for well-designed clinical trials and further insights into molecular mechanisms. Finally, studies on regimens combining vitamin D supplementation and PE should be considered the main issues in a multidimensional therapeutic approach, which is currently quite neglected. Personalized interventions based on genetics, vitamin D levels, and fitness may represent the future of prevention strategies.

Neurodegenerative diseases are expanding—also due to the growing number of elderly populations—with no definite cure still found, and represent a critical unmet need impacting medical and social areas [149].

Noticeably, neurodegeneration has recently been acknowledged as a tumor-permissive/tumor-promoting process [150]. In this context, both vitamin D (suggested long ago as a cancer-limiting agent in immune and cancer cell proliferation/invasion/differentiation [151,152,153,154]) and PE (an emerging discipline in cancer care, noted as exercise oncology [155,156,157]) can be considered as mechanistically convergent against cancer secondary to neurodegenerative diseases.

In this scenario, the opportunity to think of vitamin D combined with PE in the treatment to reduce neurological illness burden seems highly attractive. 

## Figures and Tables

**Figure 1 molecules-30-03823-f001:**
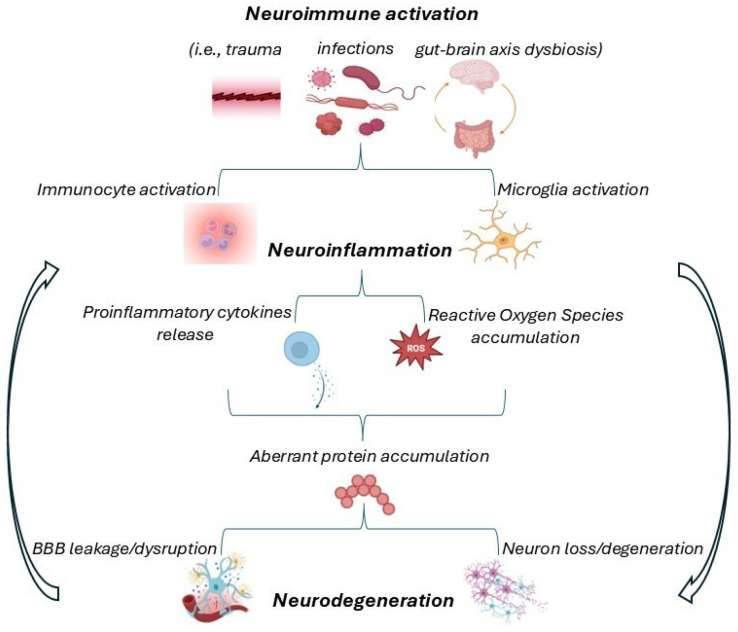
Schematic representation of the mutual loop between neuroinflammation and neurodegeneration. The initial steps trigger a proinflammatory signaling cascade that leads to neurodegenerative processes, which contribute to and perpetuate neuroinflammation.

**Figure 2 molecules-30-03823-f002:**
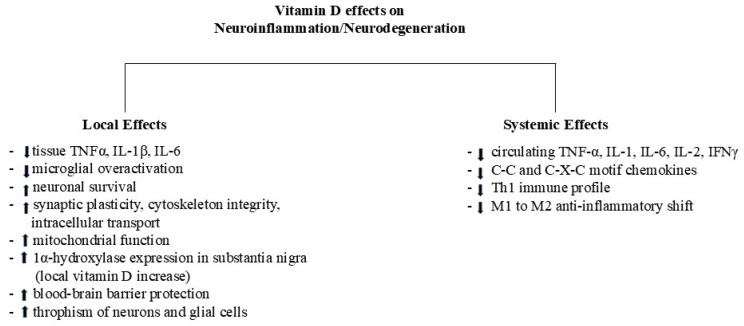
Vitamin D status drives toward neuroinflammation or neuroprotection. Adequate levels of vitamin D result in an overall anti-inflammatory effect, targeting both local and systemic mechanisms involved in neuroinflammation and neurodegeneration; up and down arrows indicate up- and downregulation, respectively.

**Table 1 molecules-30-03823-t001:** Common Targets of Physical Exercise and Vitamin D.

Targets/Pathways	PE Effects	Vitamin D Effects	Synergistic Effects
**Myokines**	↑ BDNF, ↑ Irisin, ↑ IL-6, ↑ Cathepsin B, ↑ IGF-1 [125,126] ↑ Plasticity, gray matter volume [113], anti-inflammatory effects [114]	↑ BDNF [127,128], ↑ Irisin [131], ↑ IL-6 [132,133], ↑ Cathepsin B [134], ↑ IGF-1 [135] ↓ molecular derangements [127,128] ↑ neurogeneration and neuroprotection [127,128]	↑ Microglial shift M1→M2 [129,130] ↓ IL-1β, TNFα, CN Sinflammation [129,130] ↑ neuronal sensitivity via exercise-induced ↑ VDR/Vitamin D [136]
**VDR**	↑ VDR in the brain (hippocampus, PFC) [136]	↑ VDR expression in cognitive areas [136]	↑ neuronal sensitivity to vitamin D and PE-induced effects [136]
**NF-κB**	↓ proinflammatory signaling [137]	↓ inflammatory gene expression [137]	↓ TNFα, CNS inflammation [137]
**Nrf2**	↑ antioxidant enzymes [138,139]	↑ HO-1, GPX4 [139] ↓ ferroptosis [139]	↓ oxidative and neuroinflammatory stress [138,139]
**Proinflammatory cytokines**	↓ via immunoregulation [117,118,119] ↑ myokines [120,121,122,123]	↓ via NF-κB [137], ↑ Nrf2 [138,139]	↓ TNFα, IL-1β [129,130]
**Plasticity/Cerebral blood flow**	↑ plasticity and flow [111,112]	↑ VDR-supported [136]	↑ multidimensional neuroprotective effects [140]

Molecular and signaling paths as common targets modulated by vitamin D and PE, potentially resulting in a synergistic effect; up and down arrows indicate up- and downregulation, respectively. Abbreviations are reported. PE: Physical Exercise; BDNF: Brain-Derived Neurotrophic Factor; IL-6: Interleukin-6; IGF-1: Insuline-like Growth Factor 1; VDR: Vitamin D Receptor; IL-1β: Interleukin-1 beta; TNFα: Tumor Necrosis Factor alpha; CNS: Central Nervous System; NF-κB: Nuclear Factor-kB; Nrf2: Nuclear Factor Erythroid 2-Related Factor 2; HO-1: Heme Oxygenase 1; GPX4: Glutathione Peroxidase 4; M1/M2: Microglial phenotypes: M1 (proinflammatory), M2 (anti-inflammatory); PFC: Prefrontal Cortex.

## Data Availability

No new data were created or analyzed in this study. Data sharing is not applicable to this article.

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
