# Peer review of "Vitamin D Associated with Exercise Can Be Used as a Promising Tool in Neurodegenerative Disease Protection"

_molecules, 2025, doi:10.3390/molecules30183823_

Round 1

Reviewer 1 Report

Comments and Suggestions for Authors

The article aims to describe the neuroprotective action of vitamin D in neurodegenerative processes. This paper addresses an interesting and timely issue. However, the approach does not appear to be sufficiently systematic, which impacts the results and conclusions. The search strategy is incomplete (or possibly insufficiently described). The results do not mention how many papers were identified, how duplicates were eliminated and how papers were selected.

               The review currently lacks a clear thesis or novel insight that distinguishes it from existing literature. While many established concepts are well summarized, the manuscript would benefit from a more focused narrative that offers a unique synthesis or conceptual framework.

               The article's is poorly structured; the text is fragmented, requiring fluidity in writing and connection between points.

               The article uses the keyword "exercise," and the topic "This narrative review aims to describe vitamin D as an optimal neuroprotective molecule, potentially with higher efficacy when combined with physical exercise". A suitable title would be helpful.

Table 1 is actually a table, adapt to the table model.

Improve the quality of figures               

I recommend a major revision that clarifies the manuscript's contribution, narrows its focus, improves structural flow, and addresses the citation and language issues. With these improvements, your work has the potential to make a meaningful contribution to the field.

Author Response

- The search strategy is incomplete (or possibly insufficiently described). The results do not mention how many papers were identified, how duplicates were eliminated and how papers were selected.

R:We thank R1 for appreciating the topic of the review.

For paper selection we used mainly Pubmed and Google Scholar. We did not use software tools to perform paper screening (and duplicate elimination), i.e., Covidence.

The mesh terms used were:

  • Vitamin D and Neurodegeneration:

(“Vitamin D” OR “Vitamin D Receptor” OR  “Calcitriol” OR “25-Hydroxyvitamin D”) AND (“Neurodegenerative Diseases” OR “Neurodegeneration” OR “Neuroinflammation” OR “Alzheimer Disease” OR “Parkinson Disease” OR “Multiple Sclerosis” OR “cognitive function” OR “Amyotrophic Lateral Sclerosis” OR Hungtington’s Disease” )

  • Vitamin D and Neuroinflammation:

(“Vitamin D Receptor” OR “Vitamin D” OR “Calcitriol” OR “25-Hydroxyvitamin D”) AND (“Neuroinflammation” OR “Neuroinflammatory” OR “Microglia” OR “Cytokines”)

  • Vitamin D and exercise and Neuroinflammation:

(“Vitamin D” OR “Vitamin D supplementation”) AND (“Physical Exercise” OR “Physical Activity” OR “Aerobic Exercise” OR “Resistance training”) AND (“Neuroprotection” OR “Cognition Disorders” OR “Neuroinflammation” OR “Neurodegeneration”)

Combining the mesh terms, the papers were selected based on the following criteria.

Inclusion criteria:

  • recent peer-reviewed original publications, including articles, systematic reviews, meta-analyses, reports/trials with clearly defined end-points
  • human or experimental investigations strictly focused on neurodegenerative diseases, PD, AD, ALS, HD, MS related to vitamin D status, deficiency/sufficiency or supplementation and/or to physically sedentary or active habits
  • human or experimental investigations on biomolecular mechanisms in neuroinflammatory/neurodegenerative diseases related to vitamin D and/or to physically sedentary or active habits

Exclusion criteria:

  • studies on mental/psychiatric diseases (i.e., autism, schizophrenia) related to vitamin D status, editorials, abstracts, paper lacking full data, duplicate reports.

We did not write the search strategy in the text since this is a narrative and not a systematic review. For R1 convenience, we include herein the method of  search, but if R1 retains it necessary, we can include it in the text.

- The review currently lacks a clear thesis or novel insight that distinguishes it from existing literature.

 R:We understand this point. The revised review focuses on the combination exercise-vitamin D potentially resulting in a multiplicative effect against neuroinflammation/neurodegeneration. Thus, we suggest including it in the multidimension therapeutic approach to neurodegenerative diseases, as stated in Introduction, lines 65-67, page 2; in paragraph 4, lines 355-357, page 10;  in Conclusion, line 473-475, page 13. To our knowledge, this combined regimen suggested as part of the therapy is a novel suggested issue in literature.

Furthermore, the revised text clearly highlights the ability of vitamin D to target the vicious local-systemic loop within neuroinflammation/neurodegeneration, which needs to be targeted ASAP, since it quickly drives toward neurodegenerating diseases, as stated in Introduction, lines 60-62; paragraph 4, lines 267-269, page 7, 281-284, page 8.

To our knowledge the importance of  systemic-local vicious loop as a target of vitamin D is quite neglected in literature, that indeed plenty describes the action of vitamin D at several separate levels. Section 3.1 of the revised manuscript is dedicated to  this important topic.

The manuscript has been totally re-organized and re-written to offer a synthetic framework and overcome defects in structure and in connection between points. Hopefully, the reading of the revised text is now  fluid and easy to follow.

-The article uses the keyword "exercise," and the topic "This narrative review aims to describe vitamin D as an optimal neuroprotective molecule, potentially with higher efficacy when combined with physical exercise". A suitable title would be helpful.

R:We appreciate R1 suggestion, and the title has been changed in “Vitamin D associated with exercise can be used as a promising tool in neurodegenerative disease protection.”.

-Table 1 is actually a table, adapt to the table model.

R: Table 1 is in word editable format as required by instructions for authors. Please  note that little changes are present in revised  Table 1 (i.e., the word “myokines”, editing issues)

-Improve the quality of figures

R: The quality of figures is improved (from 600 to 1200 dpi).

- I recommend a major revision that clarifies the manuscript's contribution, narrows its focus, improves structural flow, and addresses the citation and language issues. With these improvements, your work has the potential to make a meaningful contribution to the field.

R: Thanks to R1 for raising the critical points which help  us to ameliorate the review; now the focus is narrowed onto  vitamin D/exercise combination as an emerging tool to counteract neuroinflammation and control neurodegeneration; language issues have been revised; manuscript’s contribution is better defined.

Reviewer 2 Report

Comments and Suggestions for Authors

The manuscript “ Vitamin D can be used as a promising tool in neurodegenera-2 tive disease protection” is a hot topic and I can have a promising advance in the next years.

However, I  highly recommend that authors review the article's structure. The information in the article is relevant, but it needs to be better integrated.

Also, I suggest that in page 6, paragraph of lines 244-249,  it should include the references that support the information provided.

Also, page 6, lines 250-253, this paragraph does not have any reference that support the information provided.

Include more recent references such as:

Al-Kuraishy HM, Al-Gareeb AI, Selim HM, Alexiou A, Papadakis M, Negm WA, Batiha GE. Does vitamin D protect or treat Parkinson's disease? A narrative review. Naunyn Schmiedebergs Arch Pharmacol. 2024 Jan;397(1):33-40. doi: 10.1007/s00210-023-02656-6. Epub 2023 Aug 9. PMID: 37555855; PMCID: PMC10771600.

Pal R, Choudhury S, Kumar H, Dey S, Das N, Basu BR. Vitamin D deficiency and genetic polymorphisms of vitamin D-associated genes in Parkinson's disease. Eur J Neurosci. 2023 Sep;58(5):3362-3377. doi: 10.1111/ejn.16098. Epub 2023 Jul 24. PMID: 37485791.

Mirarchi A, Albi E, Beccari T, Arcuri C. Microglia and Brain Disorders: The Role of Vitamin D and Its Receptor. Int J Mol Sci. 2023 Jul 25;24(15):11892. doi: 10.3390/ijms241511892. PMID: 37569267; PMCID: PMC10419106.

Author Response

- However, I  highly recommend that authors review the article's structure. The information in the article is relevant, but it needs to be better integrated.

R: The structure of the manuscript has been totally re-organized re-written with better integration, either within the single paragraphs or in paragraph sequence.

- Also, I suggest that in page 6, paragraph of lines 244-249,  it should include the references that support the information provided. Also, page 6, lines 250-253, this paragraph does not have any reference that support the information provided. Include more recent references such as:

Al-Kuraishy HM, Al-Gareeb AI, Selim HM, Alexiou A, Papadakis M, Negm WA, Batiha GE. Does vitamin D protect or treat Parkinson's disease? A narrative review. Naunyn Schmiedebergs Arch Pharmacol. 2024 Jan;397(1):33-40. doi: 10.1007/s00210-023-02656-6. Epub 2023 Aug 9. PMID: 37555855; PMCID: PMC10771600.

Pal R, Choudhury S, Kumar H, Dey S, Das N, Basu BR. Vitamin D deficiency and genetic polymorphisms of vitamin D-associated genes in Parkinson's disease. Eur J Neurosci. 2023 Sep;58(5):3362-3377. doi: 10.1111/ejn.16098. Epub 2023 Jul 24. PMID: 37485791.

Mirarchi A, Albi E, Beccari T, Arcuri C. Microglia and Brain Disorders: The Role of Vitamin D and Its Receptor. Int J Mol Sci. 2023 Jul 25;24(15):11892. doi: 10.3390/ijms241511892. PMID: 37569267; PMCID: PMC10419106.

R: The revised text is improved with recent papers and new references quoted. References 70-78 are quoted in the revised text, lines 263-276, pages 7-8, corresponding to ex 244-253, page 6 in the original manuscript.

We did include the suggested references, quoted as numbers  55, 73 and 74 in the revised text. The number of references shifted  from 111 to 147.

Reviewer 3 Report

Comments and Suggestions for Authors

The paper entitled 'Vitamin D can be used as a promising tool in neurodegenerative disease protection' is well-written and has great potential to be published in Molecules. Authors emphasize that Vitamin D may help protect the brain in diseases like Parkinson's, Alzheimer's, Multiple Sclerosis, and Huntington's. Vitamin D works in different ways to reduce brain inflammation, which is important in these conditions. Low vitamin D levels are linked to a higher risk of these diseases. Exercise might make vitamin D work better. However, studies show mixed results, so this review might pave the way for future studies in this field. I have major criticism for this work: 

1) Section 2.1 doesn't even mention info regarding Vitamin D and has very basic, well-known information. The author needs to remove this section or modify it by adding more information regarding the mechanisms of vitamin D deficiency in inflammation and neurodegeneration.
2)The authors need to discuss the role of vitamin D and exercise in testosterone production, as this important mechanism is currently missing.
3) The authors need to add a table listing myokines that can be induced by vitamin D and physical activity. 

Author Response

1) Section 2.1 doesn't even mention info regarding Vitamin D and has very basic, well-known information. The author needs to remove this section or modify it by adding more information regarding the mechanisms of vitamin D deficiency in inflammation and neurodegeneration.

R: We are grateful to R3 for the appreciation of our work and the precious suggestion to improve the manuscript.

Section 2.1, in the revised text section 3.1, wants to focus onto the vicious loop existing between local and systemic areas during neuroinflammation and neurodegeneration.

This detrimental loop deserves attention because it represents a potential target of vitamin D: importantly, as soon as it is counteracted as better results the neuroprotection.  This suggestion is in line with the observation reported in literature (please see lines 265-269, page 7, in the revised text). 

The mechanisms of vitamin D against neuroinflammation and neurodegeneration, are addressed in section 4, entitled  “Vitamin D status: towards neuroprotection or neurodegeneration”, that is entirely reorganized, dedicated to vitamin D effects, enriched with the description of additional biomolecular paths as vitamin D targets,  topics on  vitamin D status/deficiency and  related mechanisms in neurodegenerative diseases (lines 205-217, page 6; 235-237, 242-246, page 7; 274-276, page 8; 309-311, 315-317, page 9; 386-389, 394-395, page 11). Related references are quoted. The number of references shifted  from 111 to 147.

2) The authors need to discuss the role of vitamin D and exercise in testosterone production, as this important mechanism is currently missing.

R: The issue on testosterone production, vitamin D and exercise is addressed in the revised text, paragraph 5, lines 432-449, pages 12-13.

3)  The authors need to add a table listing myokines that can be induced by vitamin D and physical activity. 

R: The first line of Table 1 reports the myokines upregulated by both PE and vitamin D. These myokines represent common targets through which the synergy might occur, among the other actions summarized in the table. To make this issue clearer, we add the word “myokines” in the table. Comments on the upregulation induced by vitamin D of myokines crossing BBB are addressed in paragraph 5, lines 386-389 and 394-395, pages 10-11.

Round 2

Reviewer 1 Report

Comments and Suggestions for Authors The article presents improvements in its construction. I suggest improving the quality of the figure.

Author Response

Comment: The article presents improvements in its construction. I suggest improving the quality of the figure.

Response: We used a professional program for figures which have been saved and replaced in text in tiff format. The quality is improved to 1500 dpi. Hopefully, now the figures are plenty legible.

Please note that the conclusion of the revised text  addresses a topic on the association neurodegenerative disease-cancer development, as recently updated in literature; in this context, we further sustain the importance of vitamin D and PE combination as potential intervention mechanistically convergent against cancer secondary to neurodegenerative diseases. Reference list is updated accordingly.

The changes are tracked in the revised text.

Reviewer 3 Report

Comments and Suggestions for Authors

It is not clear from Table 1 which myokines are upregulated. The authors need to list them in the appropriate columns.

Author Response

Comment: It is not clear from Table 1 which myokines are upregulated. The authors need to list them in the appropriate columns.

Response: Table 1 is re-organized and updated; in the revised Table 1 PE-  and vitamin D-induced  upregulation of myokines is indicated with up arrows; references are updated.

Please note that the conclusion of the revised text  addresses a topic on the association neurodegenerative disease-cancer development, as recently updated in literature; in this context, we further sustain the importance of vitamin D and PE combination as potential intervention mechanistically convergent against cancer secondary to neurodegenerative diseases. Reference list is updated accordingly.

The changes are tracked in the revised text.